# Renoprotective Effects of Tanshinone IIA: A Literature Review

**DOI:** 10.3390/molecules28041990

**Published:** 2023-02-20

**Authors:** Zhengtao Chen, Haoyue Feng, Chuan Peng, Zehua Zhang, Qianghua Yuan, Hong Gao, Shiyun Tang, Chunguang Xie

**Affiliations:** 1Clinical Medical College, Chengdu University of Traditional Chinese Medicine, Chengdu 610075, China; 2TCM Regulating Metabolic Diseases Key Laboratory of Sichuan Province, Chengdu 610075, China

**Keywords:** tanshinone IIA, natural product, renal diseases

## Abstract

The kidney is an important organ in the human body, with functions such as urine production, the excretion of metabolic waste, the regulation of water, electrolyte and acid–base balance and endocrine release. The morbidity and mortality of kidney diseases are increasing year by year worldwide, and they have become a serious public health problem. In recent years, natural products derived from fungi, plants and animals have become an important alternative source of treatment for kidney diseases because of their multiple pathways, multiple targets, safety, low toxicity and few side effects. Tanshinone IIA (Tan IIA) is a lipid-soluble diterpene quinone isolated from the Chinese herb *Salvia miltiorrhiza*, considered as a common drug for the treatment of cardiovascular diseases. As researchers around the world continue to explore its unknown biological activities, it has also been found to have a wide range of biological effects, such as anti-cancer, anti-oxidative stress, anti-inflammatory, anti-fibrotic, and hepatoprotective effects, among others. In recent years, many studies have elaborated on its renoprotective effects in various renal diseases, including diabetic nephropathy (DN), renal fibrosis (RF), uric acid nephropathy (UAN), renal cell carcinoma (RCC) and drug-induced kidney injury caused by cisplatin, vancomycin and acetaminophen (APAP). These effects imply that Tan IIA may be a promising drug to use against renal diseases. This article provides a comprehensive review of the pharmacological mechanisms of Tan IIA in the treatment of various renal diseases, and it provides some references for further research and clinical application of Tan IIA in renal diseases.

## 1. Introduction

### 1.1. Kidney Diseases

The kidney is an important organ in the human body, with the functions of producing urine, excreting metabolic waste, regulating water, electrolyte and acid–base balance and releasing endocrines. Kidney diseases are primary or secondary diseases that occur in the glomerulus, tubules, interstitium and renal blood vessels and can be divided into acute kidney injury (AKI) and chronic kidney disease (CKD), depending on the course of the disease. Common kidney diseases include glomerulonephritis, nephrotic syndrome (NS), diabetic nephropathy (DN), uric acid nephropathy (UAN), renal cell carcinoma (RCC), renal tubular acidosis, interstitial nephritis, renal artery stenosis, etc. Due to their high prevalence and poor long-term clinical prognosis, they have become a major cause of morbidity and mortality and a serious public health problem worldwide. Globally, an estimated 850 million people experience some form of kidney disease, and it has become the 11th leading cause of death worldwide [1]. Despite tremendous efforts, the current state of treatment for kidney disease is not promising. Although several drugs have been used to prevent and treat kidney disease, such as renin–angiotensin–aldosterone system (RASS) inhibitors, vasopressin receptor antagonists (VRA) and sodium-glucose transporter 2 inhibitors (SGLT2i) [2,3,4], they are still not effective in preventing the majority of patients from progressing to end-stage renal disease (ESRD) nor are they effective in helping patients move off renal replacement therapy (e.g., maintenance dialysis or renal transplantation). Dialysis is not considered an appropriate treatment option because it leads to significant medical expenditure and negatively affects patients’ quality of life [5,6]. In addition, although kidney transplantation allows patients to restore renal function, there is often a shortage of organ donors and a risk of post-transplant organ rejection. Therefore, the search for emerging and effective therapeutic options that can effectively combat kidney disease as well as improve the survival and quality of life of patients with kidney disease must continue. In recent years, natural products from fungi, plants and animals for medical treatment have emerged, and there is growing evidence that natural products with multiple pathways, multiple targets, safety, low toxicity and few side effects are important alternative therapeutic sources for the treatment of kidney diseases, meaning they have become a hot spot for the development of relevant new drugs [7].

### 1.2. Overview of Tanshinone IIA

Danshen, mainly produced in Sichuan, Shanxi, Hebei, Jiangsu and Anhui provinces in China, is the dried rhizome of *Salvia miltiorrhiza*, a genus of the family Labiatae, which has been used for thousands of years in China. The book *Shennong Ben Cao Jing* first recorded Danshen, which is bitter in taste, slightly cold in nature, enters the heart, pericardium and kidney meridians and has the effects of activating blood circulation, regulating menstruation, resolving blood stasis, relieving pain, cooling the blood, eliminating carbuncles and calming the mind [8]. In Asia, it has been widely used for centuries in the treatment of cardiovascular and cerebrovascular diseases. Since the early 1930s, the chemical composition and biological activity of *Salvia miltiorrhiza* have been studied in depth. Its chemical composition mainly consists of lipid-soluble diterpene quinones and water-soluble phenolic acid components. Tanshinone IIA (Tan IIA), a fat-soluble diterpene quinone, is the most abundant active ingredient in *Salvia miltiorrhiza* [9], which was first isolated by Nakao, a Japanese scholar, in 1934, and its chemical structure was identified as a representative monomeric compound [10]. A number of studies have described its promising protective effects against cardiovascular disease [11,12,13]. Because of the quinone-type characteristic structure in the molecule of Tan IIA, it has a more active electronic behavior and can easily undergo oxidation–reduction reactions and can participate in many biochemical reactions in the body with various biological activities, including antioxidant [14], anti-inflammatory [15], anti-fibrotic [16], antiviral [17], anti-tumor [18], anti-platelet aggregation [19], neuroprotective [8] and hepatoprotective activities, etc. [20]. 

Figure 1A shows a picture of *Salvia miltiorrhiza*, and Figure 1B shows the chemical structure of Tan IIA (C19H18O3, from PubChem CID:164676 Molecular Formula:C19H18O3,1,6,6-trimethyl-6,7,8,9-tetrahydrophenanthro [1,2-b]furan-10,11-dione, MW:294.33). Its core structure contains four rings, including naphthalene or tetrahydronaphthalene rings, an o- or p-naphthoquinone or lactone ring and a furan or dihydrofuran ring. Tan IIA is a fat-soluble cherry red needle crystal, slightly soluble in water and easily soluble in ether, acetone, ethanol, benzene and other organic solvents. Its ethanol solution and aqueous solution decrease in stability with increasing temperature, and with a melting point of 209~210°C, Tan IIA is unstable. The stability of Tan IIA is mainly influenced by temperature and light, and when the temperature is higher than 85°C, it is easily degraded by heat. Therefore, the finished product of Tan IIA should be stored in brown bottles away from light. Due to the low water solubility and low bioavailability of Tan IIA, it has been prepared in injectable emulsions, micelles and solid lipid nanoparticles to improve bioavailability [21]. In addition, the water-soluble derivative of Tan IIA, STS, prepared by adding a sodium sulfonate to the dihydrofuran ring at the C-16 position of Tan IIA, see Figure 1C, has better polarity and water solubility and bioavailability, and STS is often used as a substitute for Tan IIA and has been used interchangeably in many previous studies. In addition, clinically, STS has been approved by the China Food and Drug Administration (CFDA) for the treatment of cardiovascular diseases. The pharmacokinetics of Tan IIA and STS have been fully described in a number of studies [12,20,22].

In recent years, a number of studies have elaborated on the renoprotective effects of Tan IIA or STS in a variety of renal diseases, including DN, RF, UAN, RCC and drug-induced kidney injury caused by cisplatin, vancomycin and acetaminophen (APAP). These effects imply that Tan IIA may be a promising drug against renal diseases. In this article, we highlight the interventional effects of Tan IIA in various renal diseases in order to inform further research and clinical treatment in renal diseases.

## 2. Renal Protective Effects of Tan IIA

In recent years, an increasing number of preclinical studies have demonstrated the beneficial role of Tan IIA in the management of various renal diseases. This article systematically describes the role of Tan IIA in various renal diseases and presents all available in vivo and in vitro research results, which are presented in a tabular format (Table 1) in addition to the text to facilitate the reader’s access to information.

### 2.1. Diabetic Nephropathy

DN is a clinical syndrome characterized by glomerular hyperfiltration, a progressive increase in the urinary albumin excretion rate and a persistent decrease in the glomerular filtration rate (GFR). It is one of the most serious microvascular complications of diabetes mellitus (DM). Pathophysiological changes in DN include inflammatory cell infiltration, tubular and glomerular hypertrophy, mesangial expansion, fibrosis, extracellular matrix accumulation, cell dysfunction and death [23,24,25,26]. About 20–40% of DM patients will develop DN [27]. In addition, DN is the main cause of ESRD, and about 30–50% of ESRD cases are caused by DN [28], which causes a heavy burden on global public health. A number of studies have described the therapeutic effects of Tan IIA on DN, which may be a promising drug for the treatment of DN. Chen et al. [29] found that Tan IIA was able to reduce proteinuria and KWI (kidney weight index), reduce renal histopathological damage, reduce malonyldialdehyde (MDA) levels and increase superoxide dismutase (SOD) levels in the renal tissues of streptozocin (STZ)-induced DM rats without affecting blood glucose and glycated hemoglobin, while it reduced the levels of inflammatory and pro-fibrotic factors, including monocyte chemoattractant protein-1 (MCP-1), tissue transforming growth factor-β1 (TGF-β1), P-selectin and C-reactive protein (CRP) in the renal cortex and serum, indicating that Tan IIA could exert protective effects against early kidney injury in DM rats through anti-inflammatory, anti-fibrotic and anti-oxidative stress activities. Chen et al. [30] found that Tan IIA could enhance the proliferation of rat mesangial cells (HBZY-1) induced by high glucose in vitro, while reducing TGFβ and p65 mRNA expression. In addition, treatment with siTGFβ or sip65 also inhibited the proliferation of HBZY-1 cells. Tan IIA can reduce renal hypertrophy and 24 h urinary protein excretion in diabetic rats induced by STZ combined with a high-fat diet to a certain extent, and the expression of TGFβ and p65 protein in the kidney tissue of DM rats is also significantly reduced. It is concluded that Tan IIA may improve DN by down-regulating the TGFβ/p65 pathway. Xu et al. [31] found that Tan IIA was able to reduce blood glucose, body weight and renal function indices (uric acid (UA) and blood urea nitrogen (BUN)) in STZ-induced DM rats and attenuate collagen formation, a major component of the extracellular matrix (ECM) in the tubular lumen and glomeruli, thereby inhibiting glomerular hypertrophy and tubular lumen narrowing. Mechanistic studies revealed that Tan IIA was able to elevate the level of SOD, inhibit the high expression of phosphorylated protein kinase R-like endoplasmic reticulum kinases (p-PERK), phosphorylated eukaryotic initiation factor 2α (p-eIF2α) and activating transcription factor-4 (ATF4), key proteins of the PERK pathway (but not activator of transcription 6 (ATF6) and inositol-dependent enzyme 1 (IRE1) pathways) in the endoplasmic reticulum stress (ERS)-related pathway, and reduce the levels of TGF-β1, thrombospondin-1 (TSP1), CCAAT/enhancer binding protein homologous protein (CHOP) and glucose-regulating protein 78 (Grp78) in renal tissue mRNA, suggesting that Tan IIA may inhibit oxidative stress by increasing SOD activity, thereby inhibiting the PERK pathway in ERS, leading to TGFβ1/TSP-1 signaling pathway inhibition, ultimately reducing collagen deposition and improving DN. Li et al. [32] used bioinformatics to find that TGF-β1 may be the core target of Tan IIA for DN treatment. Further study found that Tan IIA could significantly down-regulate the expression level of TGF-β1 in high glucose (HG)-induced human kidney-2 (HK-2) cells, reduce cell death and inhibit the mRNA expression of inflammatory factors tumor necrosis factor-α (TNF-α) and interleukin-6 (IL-6). In addition, caspase-3, caspase-9, GRP78, CHOP and cleaved-caspase-12 and other apoptosis-related proteins did not change significantly. However, it could reduce the mRNA expression of pyroptosis-related interleukin-18 (IL-18) and interleukin-1β (IL-1β) and the protein expression of matured IL-1β, cleaved-caspase-1 and N-gasdermin D (GSDMD) in a concentration-dependent manner. This suggests that Tan IIA inhibits HG-induced human kidney-2 (HK-2) cell damage by blocking pyroptosis rather than apoptosis, and this effect may be related to the downregulation of TGF-β1 expression. Further rescue experiments verified that knockdown of TGF-β1 further reduced the expression of ECM-related genes (fibronectin (FN) and type Ⅰ collagen (COL-Ⅰ)) and nuclear factor-κB (NF-κB) pathway-related genes (p65, p-I-κB kinase β (IKKβ), p-nuclear factor κB inhibitor protein α (IκBα), inflammatory factors and pyroptosis-related proteins. Overexpression of TGFβ1 could significantly reverse the effect of Tan IIA, suggesting that Tan IIA could inhibit HG-induced inflammation and pyroptosis in HK-2 cells by down-regulating TGFβ1. Chen et al. [33] found that STS could significantly reduce triglyceride (TC), total cholesterol (TG), serum creatinine (Cr), blood urea nitrogen (BUN) and the urinary albumin-to-creatinine ratio (ACR) in STZ-induced DM rats, showing its good renal protection and blood lipid regulation effects in DM rats. Further studies showed that STS could increase the activities of SOD and glutathion peroxidase (GSH-Px) and decrease the activity of MDA in the renal cortex. At the same time, STS could upregulate the target SIRT1, which is closely related to DM and its complications, and downregulate the acetylated forkhead box transcription factor (FoxO1) protein in the kidney. The results suggest that STS may protect endothelial cells from DN by inhibiting oxidative stress and regulating SIRT1 and the acetylation of FoxO1 protein. Cai et al. [34] found that STS significantly reduced advanced glycation end product (AGE)-induced advanced glycosylation end-product receptor (RAGE) mRNA expression in human mesangial cells (HMCs). Meanwhile, the levels of the oxidative stress markers SOD, GSH-Px and MDA were reduced, suggesting that Tan IIA may exert its protective effect against DN through the RAGE system.

### 2.2. Renal Fibrosis

Renal fibrosis (RF) is the main pathological change in the process of various chronic kidney diseases (CKD) progressing to ESRD and is one of the main causes of progressive decline in renal function [35,36]. Several investigators have reported the antifibrotic effects of Tan IIA on different models of renal fibrosis. Zhao et al. [37] found that Tan IIA was able to cause HG-stimulated HK-2 cells to lose their intrinsic morphology (round or polygonal cells) and transform into long shuttle-like fibroblasts. Meanwhile, the expression levels of TGF-β1, α-smooth muscle actin (α-SMA) and FN and laminin (LN) were decreased, but the levels of E-calmodulin, mRNA and heme oxygenase-1 (HO-1) were increased, suggesting that Tan IIA inhibits high glucose-induced fibrosis in HK-2 cells by suppressing extracellular matrix deposition and epithelial–mesenchymal transition (EMT), as well as reducing oxidative stress. Cao et al. [38] showed that Tan IIA may inhibit HG-induced fibrosis in HK-2 cells by downregulating the expression of zinc finger transcription factor (Snail), FN, waveform protein, α-SMA, and E-calmodulin in HG-induced HK-2 cells, and by antagonizing high glucose-induced EMT signaling. Zeng et al. [39] used HG-stimulated HK-2 cells in vitro, and the study showed that Tan IIA was able to increase E-cadherin in cells by increasing vitamin D receptor (VDR) protein levels and inhibiting β-catenin and glycogen synthetase kinase-3β (GSK-3β) protein levels and decreased a-SMA expression, thereby attenuating EMT, suggesting that Tan IIA may inhibit the Wnt/β-catenin pathway by upregulating VDR, and thus attenuates high glucose-induced EMT in HK-2 cells. Zhang et al. [40] found that Tan IIA may promote autophagy in the renal tissues of DN rats and HK-2 cells induced by high glucose by regulating the miR-34-5p/Notch1 axis, thereby improving renal tubular fibrosis induced by DN. Wang et al. [41] found that Tan IIA reduced urinary protein excretion and serum creatinine (SCR) levels and attenuated glomerulosclerosis and renal inflammation in CKD rats. Further mechanistic studies showed that Tan IIA treatment significantly reduced the overactivation of the TGF-β/Smad2/3 signaling pathway associated with the increased expression of FN and type III collagen (COL-III) and type IV collagen (COL-IV) in the kidney. In addition, it inhibited the NF-κB signaling pathway associated with increased expression of TNF-a, MCP-1 and chemokine ligand-1 (CXCL-1), as well as reduced Smad7 expression in the kidneys of CKD rats, indicating that Tan IIA inhibited renal fibrosis and inflammation by altering the expression of the TGF-β-Smad and NF-κB pathways in the residual kidney. Jiang et al. [42] found that Tan IIA significantly attenuated the accumulation of FN and collagen in the renal tubular interstitium of a mouse model of folic acid-induced AKI and also inhibited the recruitment of CD45+ and type I collagen (COL-I)-positive fibroblasts to the kidney; further studies revealed the expression of chemokines of fibroblasts in the kidney, such as TGFβ1 and MCP-1, and significantly reduced the early stage of renal injury. The above effects of Tan IIA may be related to the inhibition of fibroblast recruitment to the kidney by inhibiting the expression of TGF-β1 and MCP-1, suggesting that Tan IIA may be a new therapy to prevent the progression of CKD after AKI. Tang et al. [43] pretreated rat renal fibroblasts (NRK/49F) with Tan IIA and found that Tan IIA pretreatment dose-dependently downregulated TGFβ1-induced FN and p-Smad2/3 expression, while inhibiting the proliferation and phenotypic transformation of renal interstitial fibroblasts, suggesting that Tan IIA may play an inhibitory role in renal interstitial fibrosis by blocking the TGFβ1/Smads signaling pathway in renal interstitial fibroblasts. Feng et al. [44] found that Tan IIA combined with γ-secretase inhibitors (DAPT) can improve the renal interstitial pathological damage of unilateral ureteral obstruction (UUO) model rats and reduce the synthesis and deposition of collagen fibers in the process of renal fibrosis. The mechanism may be related to the inhibition of the Notch intracellular domain (NICD), recombination signal binding protein-J kappa (Rbp-Jk), hairy enhancer of split related to YRPW motif family members 1 (Hey1) and Col I expression in renal tissue. The influence of Notch/Jagged signaling pathway transduction is related to this. Wang et al. [45] used intragastric administration of adenine solution to establish a rat model of chronic renal failure to study the effect of Tan IIA on pathological damage and renal fibrosis in rats with chronic renal failure. The results showed that Tan IIA could reduce 24 h urine volume and 24 h urine protein content in rats with chronic renal failure. The levels of BUN and SCr in the serum and the contents of COL-Ⅳ, type III procollagen (PC-Ⅲ) and LN in the kidney can increase the level of FN in the serum and improve the pathological damage to the kidney.

### 2.3. Uric Acid Nephropathy

Hyperuricemia (HUA) is an independent risk factor for the development and progression of chronic kidney disease, and if the blood uric acid concentration is supersaturated for a long time, it will increase the burden on the kidney and cause uric acid crystals to be deposited in the kidney, resulting in different degrees of kidney damage, which is called uric acid nephropathy (UAN) [46,47]. Wu et al. [48] found that tanshinone IIA inhibited the overexpression of MCP-1 and IL-1β in renal tissues of rats with uric acid nephropathy and inhibited the translocation of NF-κB from the cytoplasm to the nucleus, thereby inhibiting inflammation against uric acid nephropathy. Zhang et al. [49] found that Tan IIA could inhibit the expression of nicotinamide adenine dinucleotide phosphate oxidase family oxidase 4 (NOX4)/p22phox and inhibit the activation of the mitogen-activated protein kinase (MAPK) pathway in vitro and in vivo to treat UAN.

### 2.4. Renal Cell Carcinoma

Renal cell carcinoma (RCC) is a malignant tumor originating from the urinary tubular epithelial system of the renal parenchyma, called renal cell carcinoma or renal adenocarcinoma, or simply renal cancer. Renal cell carcinoma is characterized by reduced renal filtration, anemia and elevated blood pressure, ultimately leading to renal failure [50]. Wei et al. [51] found that Tan IIA induces apoptosis in 786-O cells, possibly through the activation of P53 expression and thus activation of the downstream p21-mediated cell cycle perturbation and mitochondria-mediated intrinsic cell death pathways. Kim NY et al. [52] found that Tan IIA was able to play a therapeutic role in RCC by inhibiting β-catenin translocation to the nucleus, thereby inducing autophagy-related cell death.

### 2.5. Renal Ischemia and Reperfusion Injury

Renal ischemia and reperfusion injury (RIRI) are major causes of acute renal failure and are associated with delayed recovery of graft function, rejection and chronic graft dysfunction after renal transplantation [53]. Xu et al. [54] found that Tan IIA pretreatment improved renal function in RIRI rats, and this effect was partly achieved by downregulating the expression of myeloperoxidase (MPO), inflammatory response, macrophage migration inhibitor factor (MIF), caspase-3 and p-p38 MAPK. Yang et al. [55] found that Tan IIA may have a protective effect on the hypoxia/reperfusion (H/R) of HK-2 cells by reducing apoptosis through the inhibition of the oxidative stress response. Tai et al. [56] found that Tan ⅡA pretreatment could regulate mitochondrial function through the PI3K/Akt/Bad pathway in obese rats, thereby reducing RIRI-induced apoptosis, inflammatory factors, oxidation and the antioxidant balance, improving renal function and reducing renal pathological damage in obese rats.

### 2.6. Drug-Induced Kidney Injury

#### 2.6.1. Cisplatin-Induced Renal Injury

Cisplatin is a commonly used antitumor drug with a broad spectrum of antitumor activities and high anticancer activity. Dou et al. [57] firstly identified the nuclear receptor family as a new target for cisplatin-induced AKI by network pharmacology, and further in vitro and in vivo studies revealed that Tan IIA could reduce renal inflammation and pathological damage by inhibiting the pregnane X receptor (PXR)-mediated NF-κB signaling pathway through activating PXR.

#### 2.6.2. Vancomycin-Induced Renal Injury

Vancomycin (VAN) is a glycopeptide antibiotic with powerful bactericidal effects against Gram-positive bacteria, is an important drug for the treatment of serious infections caused by methicillin-resistant *Staphylococcus aureus* (MRSA) and is one of the commonly used antibacterial drugs. Its excretion through the kidney has a certain degree of nephrotoxicity, which can lead to corresponding kidney damage. Overseas studies have shown that the incidence of VAN nephrotoxicity is about 5–25% [58]. Xi et al. [59] found that, compared with the vancomycin-induced renal injury model group, the serum levels of cystatin C (Cys C), Scr, BUN, urinary 24 h protein amount and urinary N-acetyl-B-D glucosaminidase (NAG) and kidney injury molecule 1 (KIM-1) levels, as well as the levels of MDA and nitric oxide (NO) in renal tissues, were significantly lower in the STS administration group, and the levels of SOD and GSH-Px in renal tissues were significantly higher. GSH-Px levels in renal tissues were significantly increased; the pathological changes in the kidney were significantly reduced, indicating that STS could reduce vancomycin-induced renal injury, and the mechanism may be related to the inhibition of oxidative stress.

#### 2.6.3. APAP-Induced Kidney Injury

APAP is the most widely used antipyretic and analgesic drug for headache and fever caused by influenza or cold, but it often causes kidney injury when taken in excess [60]. Zhang et al. [61] illustrated the preventive effect of Tan IIA against APAP-induced nephrotoxicity by administering Tan IIA continuously to mice for 1 week before administration of APAP. The results showed that Tan IIA pretreatment significantly attenuated the APAP-induced increase in the serum creatinine level and the associated pathological damage compared with the control group. Further mechanistic studies showed that this effect may be through the up-regulation of the Nrf2-MRP2/4 pathway to promote the clearance of toxic intermediate metabolite N-acetyl-p-benzoquinone imine (NAPQI) from the kidney.

### 2.7. Contrast-Induced Nephropathy

Contrast-induced nephropathy (CIN) refers to an acute decline in renal function within 3 days after the intravascular administration of contrast medium on the basis of exclusion of other causes. The incidence of CIN ranges from 3% to 14% and can be as high as 20% in high-risk patients [62,63]. Liang et al. [64] studied the therapeutic effect of Tan IIA on CIN in an in vivo model of ioversol-induced kidney injury in SD rats and an in vitro cell model of HK2 cells treated with oxidative stress (H2O2). They found that Tan IIA reduces tubular necrosis, apoptosis and oxidative stress in rats and alleviates oxidative stress in HK2 cells, which may be related to the activation of the nuclear factor-related factors (Nrf2)/antioxidant response element (ARE) pathway.

### 2.8. Henoch–Schonlein Purpura Nephritis

The main pathological manifestation of allergic purpura is the intravascular deposition of immune complexes, which is a common systemic small vessel disease and can be accompanied by renal damage in severe cases, even endangering the patient’s life. Henoch–Schonlein purpura nephritis (HSPN) is a secondary nephropathy, which is more common in pediatrics, and most of them have a good prognosis, while a few patients have prolonged disease and even develop chronic renal insufficiency [65]. Yang et al. [66] found that Tan IIA may exert therapeutic effects on mice with HSPN by decreasing galactosyltransferase chaperone protein (Cosmc) levels in kidney tissues, increasing advanced protein oxidation product (AOPP) levels in kidney tissues, and activating the extracellular signal-regulated kinase (ERK)/MAPK signaling pathway.

### 2.9. Hypertensive Renal Damage

Hypertensive renal damage is one of the most common complications of hypertension, and long-term sustained hypertension can cause small renal artery lesions and glomerulosclerosis, which in turn can lead to renal insufficiency and even renal failure. Liu et al. [67] found that mullein isoflavones combined with Tan IIA (compounds from Astragalus and Salvia, respectively) could intervene in Ang II-induced rat renal artery endothelial cell (RRAEC) injury by promoting endothelial cell proliferation, inhibiting apoptosis, reducing reactive oxygen species (ROS) production and increasing endothelial nitric oxide synthase (eNOS) levels, thus exerting a protective effect against hypertensive kidney injury through multiple factors. A recent study by Liu et al. [68] revealed the mechanism of Tan IIA on Ang II-induced impairment of RRAECs at the miRNA level, and they found that mullein isoflavone-Tan IIA ameliorated the Ang II-induced impairment of proliferation and the migratory function of RRAECs; this mechanism may be related to the miRNA-200c-3p/Zinc finger E-box binding homeobox 2(ZEB2) pathway.

### 2.10. Nephrotic Syndrome

Nephrotic syndrome (NS) is caused by a variety of causes, can produce a variety of pathological types and is often accompanied by a variety of complications, difficult prognosis, complex treatment procedures, unsatisfactory efficacy and may eventually develop into renal failure, placing great pressure on human life and the economy [69]. Adriamycin-induced nephropathy is recognized as a good method to simulate human nephropathy, and its clinical manifestations are similar to human nephrotic syndrome [70]. Lv et al. [71] found that STS can reduce 24-h urinary protein and blood viscosity and improve pathological tissue damage in adriamycin-induced nephropathy rats, which may be related to the inhibition of TGF-β1 and plasminogen activator inhibitor-1 (PAI-1) expression in renal tissue. Wang et al. [72] found that tanshinone IIA can increase the expression of nephrin in renal tissue but inhibit the expression of TGFβ1, reduce the excretion of urinary protein in adriamycin-induced nephropathy rats, reduce and delay glomerulosclerosis and play a role in protecting the kidney.

### 2.11. Acute Kidney Injury Induced by Sepsis

Sepsis is one of the main causes of acute kidney injury (AKI) in critically ill patients. Although the current level of diagnosis and treatment is improving, the prevalence and mortality of AKI caused by sepsis in critically ill patients are still high. Zhang et al. [73] found that tanshinone ⅡA could inhibit the RIP3/ FUNDC1 signaling pathway to reduce LPS-induced apoptosis in HK-2 cells, reduce Scr and BUN levels and alleviate the pathological damage of the proximal renal tubules in mice after LPS injection.

**Table 1 molecules-28-01990-t001:** Study characteristics of experiments in kidney disease.

Disease		Models and Modeling Methods	Research Type	Tan IIA or Derivatives Dose and Treatment Schedule	Targets	References
DN		SD rats; intraperitoneal injection of STZ (65 mg/kg.d) for 3 d	In vivo	Tan IIA; 10 mg/kg.d; p.o. for 12 weeks	↓ MDA, ↑SOD, ↓ MCP-1, ↓ TGF-β1, ↓ p-selectin, ↓ CRP in renal tissue; ↓ MCP-1, ↓ TGF-β1, ↓ p-selectin, ↓ CRP in serum	[29]
		HBZY-1 cells; HG (30 mmol/L) for 48 h; SD rats; intraperitoneal injection of STZ (45 mg/kg.d) combined with high-fat diet for 7 weeks	In vitro and in vivo	Tan IIA; 20, 40, and 80 µM for 48 h; 8 mg/kg.d, i.m. for 3 weeks	↓ TGFβ, ↓ p65 in cell; ↓ TGFβ, ↓ p65 in renal tissue	[30]
		SD rats; intraperitoneal injection of STZ (60 mg/kg) for 2 d	In vivo	Tan IIA; 2, 4, 8 mg/kg.day, i.p. for 6 weeks	↓ TGF-β1, ↓ TSP1, ↓ Grp78, ↓ CHOP, ↓ p-PERK, ↓ p-eIF2α, ↓ ATF4, ↑SOD in renal tissue	[31]
		HK-2 cells; HG (30 mmol/L) for 48 h	In vitro	Tan IIA; 1, 5, 10 μM for 24 h	↓ TGF-β1, ↓ TNF-α, ↓ IL-6, -caspase-3, -caspase-9, -GRP78, -CHOP, cleaved caspase-12, ↓ IL-18, ↓ IL-1β, ↓ matured IL-1β, ↓ cleaved-caspase-1, ↓ N-GSDMD, ↓ FN, ↓ COL-Ⅰ, ↓ p65, ↓ p-IKKβ, ↓ p-IκBαin cell	[32]
		SD rats; STZ (65 mg/kg) was injected intraperitoneally for 72 h.	In vivo	STS; 10, 20 mg/kg.day, i.p. for 12 weeks	↑ SOD, ↑ GSH-Px, ↓ MDA, ↓ ET, ↓ TXB2, ↓ 6-keto-PGF1α, ↓ ac FoxO1, ↑ SIRT1 in renal tissue	[33]
		HMCs; AGES (1, 10, 50, 100 μg/mL) for 48 h	In vitro	STS; 0, 0.1, 1.0, 5.0, 10.0 μg/mL for 48 h	↓ RAGE, ↑ SOD, ↑ GSH-Px, ↓ MDA in cells	[34]
RF		HK-2 cells; HG (30 mmol/L) for 24 h	In vitro	Tan IIA; 6 μg/mL for 24 h	↓ TGF-β1, ↓ α-SMA, ↑ E-cadherin, ↓ FN, ↓ LN, ↑ HO-1 in cells	[37]
		HK-2 cells; HG (30 mmol/L) for 48 h	In vitro	Tan IIA; 1, 10, 50 µM for 48 h	↑ E-cadherin, ↓ α-SMA, ↓ vimentin, ↓ FN, ↓ Snail in cells	[38]
		HK-2 cells; HG (30 mmol/L) for 48 h	In vitro	Tan IIA; 5 μM, 10 μM for 48 h	↓ α-SMA, ↑ E-cadherin, ↑ VDR, ↓ β-catenin, ↓ GSK-3β, in cells	[39]
		Kunming mice; intraperitoneal injection of STZ (50mg/kg.d) three times every other day; HK-2 cell; HG (30 mmol/L) for 48 h	In vitro and In vivo	Tan IIA; 5, 10, 25 mg/kg p.o. for 30d Tan IIA; 5 mg/L, 10 mg/L, 25 mg/L for 30 min	↑ LC 3BII/LC 3BI, ↓ P62↑ Beclin1, ↓ ATG7, ↓ Notch1, ↓ p-AKT, ↓ p-mTOR, ↑ PTEN, ↑ miR-34a-5p in renal tissue; ↑ LC 3BII/LC 3BI, ↓ P62↑ Beclin1, ↓ ATG7, ↓ Notch1, ↓ p-AKT, ↓ p-mTOR, ↑ PTEN, ↑ miR-34a-5p, ↓ Col- I, ↓ Col-III in cells	[40]
		SD rats; 5/6 nephrectomy for 1 week	In vivo	Tan IIA; 10 mg/kg.d, p.o. for 16 weeks	↓ FN, -COL-Ⅰ, ↓ COL-III, ↓ COL-IV, ↓ TGF-β1, ↓ TNF-α, ↓ CXCL-1, ↓ MCP-1, -RANTES, ↓ Smad3, ↓ p-Smad2/3, ↓ Smad7, -IKKβ, ↓ p-IKKβ, -IκBα, ↓ p-IκBα, -NF-κB, ↓ p-NF-κB in renal tissue	[41]
		C57BL/6 mice; single intraperitoneal injection of folic acid (250 mg/kg)	In vivo	Tan IIA; tail vein injection, 15 mg/kg.d for 2 days	↓ TGF-β1, ↓ MCP-1 in renal tissue	[42]
		NRK/49F cells; TGFβ1 (5 ng/mL) for 0, 6, 12, 24 h	In vitro	Tan IIA; 10^−6^, 10^−5^, 10^−4^ mol/L for 2 h	↓ FN, ↓ p-Smad2/3, -total Smad2, -total Smad3 in cells	[43]
		SD rats; unilateral ureteral obstruction	In vivo	Tan IIA; 25 mg/kg.d, DAPT 12 mg/kg.d, Tan IIA; 25 mg/kg.d+DAPT 12mg/kg.d for 2 weeks	↓ NICD, ↓ Hey1, ↓ Rbp-Jk, ↓ Col-Ⅰin renal tissue	[44]
		SD rats; adenine solution (250 mg/kg.d) was administered by gavage for 3 weeks.	In vivo	Tan IIA; 15.0 mg/kg.d for 4 weeks	↓ COL-Ⅳ, ↓ PC-Ⅲ, ↓ LN, ↑ FN in serum	[45]
UAN		SD rats; adenine (30 mg/kg.d) was administered by gavage for 18 d.	In vivo	Tan IIA; 3, 1.5, 0.75 g/mg/kg.d, p.o. for 23 days	↓ MCP-1, ↓ IL-1β, ↓ Inucleus NF-κB, ↑ cytosome NF-κB, in renal tissue	[48]
		Kunming mice; adenine (100 mg/kg.d) and potassium oxalate (150 mg/kg.d) for 28 d HK-2 cells; stimulation with uric acid (0.2 mg/L) for 24 h	In vitro and In vivo	Tan IIA; 4, 8 or 16 mg/kg.d, p.o. for 28 d Tan IIA; 1, 5, and 10 μM for 24 h	↓ NOX4, ↓ p22phox, ↓ iNOS, ↓ COX-2, ↓ p-p44/42, ↑ p44/42, ↑ p38, ↓ p-p38, ↓ p-JNK, ↑ JNK in renal tissue, ↓ ROS, ↓ NOX4, ↓ p22phox, ↓ p-p44/42, ↑ p44/42, ↑ p38, ↓ p-p38, ↓ p-JNK, ↑ JNK in cells	[49]
RCC		Human renal carcinoma cell line 786-O	In vitro	Tan IIA; 1, 2, 4, 8 µg/mL for 24 h	↑ p53, ↑ p21, ↑ Bax, ↑ caspase-3 in cells	[51]
		Human renal carcinoma cell line 786-O and human renal carcinoma Caki-1 cell line	In vitro	Tan IIA; 1, 5, 10 or 30 mM for 24 h	-PARP, -caspase-3, -Bcl-2, -Alix, ↑ LC3Ⅱ/LC3Ⅰ, ↑ Atg7, ↑ Beclin-2, ↑ p-Beclin-2, ↓ p62, ↓ β-catenin(NE), -β-catenin(CE) in cells	[52]
RIRI		SD rats; after resection of the right kidney, the left kidney was ligated for 25 min by non-traumatic aneurysm clipping	In vivo	Tan IIA; 25 mg/kg.d, i.p. for 10 days	↓ MPO, ↓ TNF-α, ↓ IL-6, ↓ MIF, ↓ cleaved caspase-3, ↑ Bcl-2, ↓ p-p38 MAPK in renal tissue	[54]
		HK-2 cells; the cells were cultured in a hypoxic incubator (94%N2, 5%CO2, 1%O2) for 24 h and then cultured in a carbon dioxide incubator for 6 h.	In vitro	Tan IIA; 5, 10, 20 μg/mL for 24 h	↓ ROS, ↓ Bcl-2, ↑ Bax, ↑ Cleaved Caspase-3 in cells	[55]
		SD rats; high-fat diet (HFD) feed for 8 weeks + the left renal artery was clamped for 30 min after right nephrectomy.		Tan IIA; 5 mg/kg.d, 10 mg/kg.d, and 20 mg/kg.d for 2 weeks	↓ TNF-α, ↓ IL-1β, ↑ SOD, ↓ MDA, ↓ caspase-9/3, ↓ cleaved caspase-9/3, ↓ Bax, ↑ Bcl-2, ↓ PARP, ↓ Cyt-c, ↓ ROS, ↑ PI3K, ↑ Bad, ↑ Akt, ↑ p-AKT, ↑ p-Bad, ↑ PCG-1α, ↑ Nrf1, ↑ Tfam, ↓ Mfn1, ↓ Mfn2, ↑ Drp1 in renal tissue	[56]
Drug-induced kidney injury	Cisplatin-induced renal injury	C57BL/6 mice; a single dose of cisplatin (20 mg/kg) was intraperitoneally injected.	In vivo	Tan IIA; 12.5 mg/kg.d and 25 mg/kg.d, i.p. for 3 d	↓ P105/p50, ↓ IKKβ, ↓ L-6, ↓ IL-1β, ↓ TGF-β, ↑ PXR, ↑ RXRα, ↓ pp65/p65, ↑ Cyp3a11 in renal tissue↓ TNF-α, ↓ IL-6 in serum	[57]
	Vancomycin-induced renal injury	SD rats; 200 mg/kg vancomycin by single intraperitoneal injection	In vivo	STS; 15, 30, 60 µg/kg.d, i.p. for 10 d	↑ SOD, ↑ GSH-Px, ↓ MDA, ↓ NO in renal tissue	[59]
	APAP-induced renal injury	C57BL/6J Nrf2^+/−^ mice; 200 mg/kg APAP for 3 h HK-2 cells; APAP (1 mM) for 24, 48, 72 h	In vitro and In vivo	Tan IIA; 10 mg/kg.d and 30 mg/kg.d, p.o. for 1 week Tan IIA; 2.5, 10 μM for 24, 48, 72 h	↑ Nrf2, ↑ MRP2, ↑ MRP4 in renal tissue ↑ Nrf2, ↑ MRP2, ↑ MRP, ↑ p-Nrf2 (in cell nucleus), ↑ p-Nrf2 (in cytoplasm), ↑ Nrf2(in cell nucleus) in cells	[61]
CIN		SD rats; indomethacin (10 mg/kg) was given after tail vein injection of 50 mg/kg sodium pentobarbital, followed by tail vein injection of ioversol (3 g/kg organic iodine). HK-2 cells; the cells were incubated with 500 µmol/L H2O2 for 5 min.	In vitro and In vivo	Tan IIA; 25 mg/kg i.h. single use Tan IIA; 40 µg/mL for 1 h	↓ MDA, ↓ 8-OHdg, ↑ Nrf2, ↑ HO-1 in renal tissue; ROS↓, ↑ Nrf2, ↑ HO-1, ↑ ARE in HK2 cells	[64]
HSPN		C57BL/6 mice; BSA (4 mL/kg/every other day for 8 weeks) was administrated intragastrically, LPS (0.025%LPS0.2 mL, once at 6, 8, 10 and 12 weeks) was injected into the caudal vein, and CCL4 (0.3 mL castor oil + 0.1 mLCCL4, once a week) was injected subcutaneously. The IgA nephropathy model was established at the same time during the modeling period, and the blood fever model was combined into the anaphylactoid nephritis model. The room temperature during the modeling period was 30 °C, and 25% dry ginger water (10 mL/kg, once every other day) was given from the 9th week, and the modeling was finished at the 12th week.	In vivo	Tan IIA; 25 μmol/L 200 μL, i.p. for 4 weeks	↑ AOPP, ↑ ERK, ↑ p-ERK, ↓ Cosmc in renal tissue	[66]
Hypertensive renal damage		RRAECs; AngⅡ was treated with 5 × 10^−7^ mol for 24 h.	In vitro	Isoflavone 1.5 mg/L, Tan IIA 3 mg/L, isoflavone 3 mg/L+Tan IIA 3 mg/L for 24 h	↓ ROS, ↑ eNOS in cells	[67]
		RRAECs; AngⅡ was treated with 5 × 10^−7^ mol for 24 h.	In vitro	Isoflavone 3 mg/L-Tan IIA 3 mg/L pretreated for 1 h	↓ miRNA -200c -3p, ↑ ZEB2 in cells	[68]
NS		SD rats; single tail vein injection of doxorubicin (6.5 mg/kg)	In vivo	STS; 3.5 and 7 mg/kg.d, i.p. for 2 weeks	↓ TGF-β1, ↓ PAI-1 in renal tissue	[71]
		SD rats; single tail vein injection of doxorubicin (7 mg/kg)	In vivo	Tan IIA; 0.02, 0.04 g/kg/d; p.o. for 4 weeks	↓ TGF-β1, ↑ nephrin in renal tissue	[72]
AKI induced by sepsis		C57BL/6 mice; LPS (10 mg/kg) was intraperitoneally injected. 10 mg/kg LPS was injected and stimulated for 24 h. HK-2 cell; LPS (10 μg/mL) for 24 h	In vitro and In vivo	Tan IIA; 10 mg/kg; i.p. in advance for 15 min Tan IIA; 10 mg/L pretreated for 1 h	↓ Cleaved-caspase3, ↓ RIP3, ↓ p^18^-FUNDC in renal tissue; ↓ Cleaved-caspase3, ↓ RIP3, ↓ p^18^-FUNDC in cells	[73]

Abbreviations: diabetic nephropathy, DN; renal fibrosis, RF; uric acid nephropathy, UAN; renal cell cancer, RCC; contrast-induced nephropathy, CIN; Henoch–Schonlein purpura nephritis, HSPN; nephrotic syndrome, NS; Sprague Dawley rats, SD rats; streptozocin, STZ; rat mesangial cells, HBZY-1; human kidney proximal tubular epithelial, HK2; high glucose, HG; human mesangial cells, HMCs; advanced glycation end products, AGES; rat renal fibroblast, NRK/49F; tissue transforming growth factor-β1, TGF-β; bovine serum albumin, BSA; lipopolysaccharide, LPS; carbon tetrachloride, CCL4; rat renal artery endothelial cells, RRAECs; angiotensin II, AngⅡ; zinc finger E-box binding homeobox 2, ZEB2; malonyldialdehyde, MDA; superoxide dismutase, SOD; monocyte chemoattractant protein-1, MCP-1; C-reactive protein, CRP; thrombin sensitive protein-1, TSP-1; glucose-regulating protein 78, GRP78; CCAAT/enhancer binding protein homologous protein, CHOP; protein kinase R-like endoplasmic reticulum kinase, PREK; eukaryotic initiation factor 2α, eIF2α; activating transcription factor-4, ATF-4; tumor necrosis factor-α, TNF-α; interleukin-6, IL-6; interleukin-18, IL-18; interleukin-1β, IL-1; gasdermin D, GSDMD; fibronectin, FN; type Ⅰ collagen, COL-Ⅰ; I-κB kinase β, IKKβ; nuclear factor κB inhibitor protein α, IκBα; glutathion peroxidase, GSH-Px; endothelin, ET; thromboxane B2, TXB2; 6-ketoprostacycline F1α, 6-keto-PGF1α; forkhead box transcription factor Ol, FOXO1; silent message modulator 1, SIRT1; advanced glycosylation end-product receptor, RAGE; α-smooth muscle actin, α-SMA; epithelial cadherin E, E-cadherin; laminin, LN; heme oxygenase-1, HO-1; vitamin D receptor, VDR; glycogen synthetase kinase-3β, GSK-3β; type III collagen, COL-III; type IV collagen, COL-IV; chemokine ligand-1, CXCL-1; T-cell expression and secretion factors, RANTES; nuclear factor-κB, NF-κB; Notch intracellular domain, NICD; hairy enhancer of split related to YRPW motif family members 1, Hey1; recombination signal binding protein-J kappa, Rbp-Jk; collagenous fiber, C-Ⅳ; type III procollagen, PC-Ⅲ; reduced coenzyme nicotinamide adenine dinucleotide phosphate oxidase family oxidase 4, NOX4; tissue-specific overexpression of nitric oxide synthase, iNOS; c-Jun amino terminal kinase, JNK; Bcl-2-associated X protein, BAX; myeloperoxidase, MPO; macrophage migration inhibitor factor, MIF; mitogen-activated protein kinase, MAPK; reactive oxygen species, ROS; pregnane X receptor, PXR; retinoic acid X receptor alpha, RXRα; nitric oxide, NO; 8-hydroxy-deoxyguanosine acid, 8-OHdg; nuclear factor-related factors, Nrf2; antioxidant response element, ARE; advanced protein oxidation products, AOPP; extracellular signal-regulated kinase, ERK; galactosyltransferase chaperone protein, Cosmc; endothelial nitric oxide synthase, eNOS; plasminogen activator inhibitor-1, PAI-1; sodium tanshinone IIA sulfonate, STS; zinc finger transcription factor, snail; γ-secretase inhibitors, DAPT; acute kidney injury induced by sepsis, AKI; microtubule-associated protein 1A/1B-light chain 3, LC3; autophagy-related 7, Atg7; B-cell lymphoma-2, Bcl-2; poly(ADP-ribose) polymerase, PARP; cellular extracts, CE; nuclear extracts, NE; cytochrome C, Cyt-c; phosphatidylinositol-3 kinase, PI3K; protein kinase B, Akt; peroxisome proliferator-activated receptor gamma coactivator-1α, PCG-1α; mitofusin1, Mfn1; mitofusin2, Mfn2; nucleo respiratory factor1, Nrf1; transcription factor A of mitochondria, Tfam; dynamin-related protein 1, Drp1.

## 3. Safety and Side Effects and Drug Interactions

A number of studies in recent years have elaborated on the toxic effects of Tan IIA. In vitro toxicity studies found that the cell viability of rat cardiomyocyte cells (H9c2 cells) treated with 0–10 µM Tan IIA for 24 h was not significantly different from that of normal controls [74]. However, some studies found that high concentrations of Tan IIA are toxic to human endothelial cells, and high concentrations (25 µM) of Tan IIA were reported to kill human endothelial cells within 24 h [75]. Wang et al. [76] performed a potential in vivo toxicity assessment using a zebrafish embryo model and found that Tan IIA was not teratogenic when its concentration in chorionic and dechorionized embryo groups was below 5 µM, but at high concentrations, it exhibited severe growth inhibition, developmental malformations and cardiotoxicity. In a clinical trial on the potential myocardial protective effect of STS on patients with non-ST-segment elevation acute coronary syndrome (NSTE-ACS) treated with PCI, 192 patients in the test group received STS, and 180 patients received saline, resulting in a 30-day incidence of major adverse cardiac events (MACEs) in the test and control groups of 18.8% and 27.2%, respectively, and without any harmful side effects [77]. STS injection is the only tanshinone preparation widely used in clinical practice in China. Anaphylactic shock is a rare but serious side effect, probably due to small impurities in the reagent during drug preparation, and this side effect can often be eliminated by improving the quality of the product [78,79]. There are still few studies on the adverse effects and side effects of Tan IIA, and especially little research has been reported on the toxic effects in renal diseases. In order to ensure the safety and efficacy of Tan IIA in the prevention and treatment of renal diseases and to better understand its effects, it is imperative to conduct large-scale randomized clinical trials and further basic research.

Drug interaction (DI) refers to the compounding effect of two or more drugs taken by a patient at the same time or within a certain period of time, which can lead to enhanced efficacy or reduced side effects, as well as weakened efficacy or undesirable toxic side effects. Many studies have described the interactions between Tan IIA and other drugs. For example, it has been shown that, after intravenous injection of Xiangdan (containing *Salvia miltiorrhiza* and *lignum dalbergiae odoriferae*) into rats, the metabolism and excretion of diterpene quinone were inhibited by *lignum dalbergiae odoriferae*, thereby increasing the bioavailability of Tan IIA [80]. Li et al. [81] found that Tan IIA enhanced the efficacy of chemotherapy with adriamycin in breast cancer while reducing its toxic side effects, including weight loss, bone marrow suppression, cardiotoxicity and nephrotoxicity. Zhang et al. found that Tan IIA altered the pharmacokinetic profile of APAP and its metabolites through the HOTAIR-Nrf2-MRP2/4 signaling pathway, thereby attenuating APAP-induced hepatotoxicity [82]. However, some studies have found synergistic attenuating effects between drugs. For example, STS can increase the metabolic rate of warfarin and displace WAR from the WAR–HSA complex, resulting in increased concentration of free WAR in the blood, resulting in increased anticoagulation, increased anticoagulation and/or even bleeding [83], suggesting that STS should be used with caution in combination with other drugs. However, few studies have described the interaction of Tan IIA with typical drugs with renal protective effects (such as ACEI, ARB, SGLT1, etc.), and whether they have synergistic effects or synergistic reductions still deserves further study.

## 4. Conclusions and Future Prospects

Most drugs are currently synthetic, and although these drugs have been widely used in clinical practice, these synthetic drugs are often accompanied by many adverse effects, which often limit their clinical application. Worse still, the overproduction of synthetic drugs is wasting resources, polluting the environment and damaging people’s health. In contrast, natural product of fungal, animal and plant origin often have the advantages of being safe, less toxic, economical and highly effective, providing a tremendously valuable resource for the development of our drugs.

Tan IIA is one of the lipid-soluble extracts from the traditional medicinal plant *Salvia miltiorrhiza*. With extensive and in-depth research, more and more studies have elaborated the preventive and curative effects of Tan IIA in various in vivo and in vitro renal disease models. In general, Tan IIA can reduce oxidative stress and inflammatory cytokine levels, anti-apoptosis, anti-fibrosis, reduce ERS, promote autophagy and reduce extracellular matrix deposition and EMT, thereby reducing renal pathological damage and improving renal function (Figure 2). However, studies on the renoprotective effects of Tan IIA still leave room for improvement as follows: (1) The exact mechanism of Tan IIA in the treatment of kidney diseases remains to be further elucidated. Many studies have reported that Tan IIA has regulatory effects on non-coding RNAs [84], RNA methylation [85], immune-related factors [86] and ferroptosis [87], and the above mechanisms have also been reported to be crucial in the occurrence and development of kidney diseases. In the future, more attention may be paid to the regulatory mechanism of Tan IIA in kidney diseases. (2) In existing studies, the types of models are still relatively limited, mainly focusing on DN and RF, while there is a lack of research on some common clinical kidney diseases, such as IgA nephropathy and chronic glomerulonephritis (CGN), and it is necessary to pay sufficient attention to this area in the future. (3) Despite growing preclinical evidence of the beneficial effects of Tan IIA in renal disease, only a few relevant randomized controlled clinical trials are available in China [88,89], and the literature on the clinical application potential of Tan IIA for the treatment of renal disease remains inadequate; large-scale randomized clinical trials and further scientific studies are necessary to ensure the safety, efficacy and better understanding of the effects of Tan IIA in order to obtain a higher level of evidence-based medicine. (4) The efficacy and mechanism of the combination of Tan IIA and related drugs or natural products commonly used in the treatment of renal diseases are still unclear, and there is still a need to improve relevant high-quality clinical and basic studies to support the use of Tan IIA as an alternative drug for the treatment of renal diseases. (5) Due to the pharmacokinetic characteristics of Tan IIA, it is meaningful to increase its aqueous solubility and bioavailability by discovering good drug carriers and effective dosage forms and to increase its distribution concentration and retention time in renal tissues.

Although there are still many problems to be solved, it is believed that, with further research, Tan IIA will become a promising drug for the prevention and treatment of kidney diseases.

## Figures and Tables

**Figure 1 molecules-28-01990-f001:**
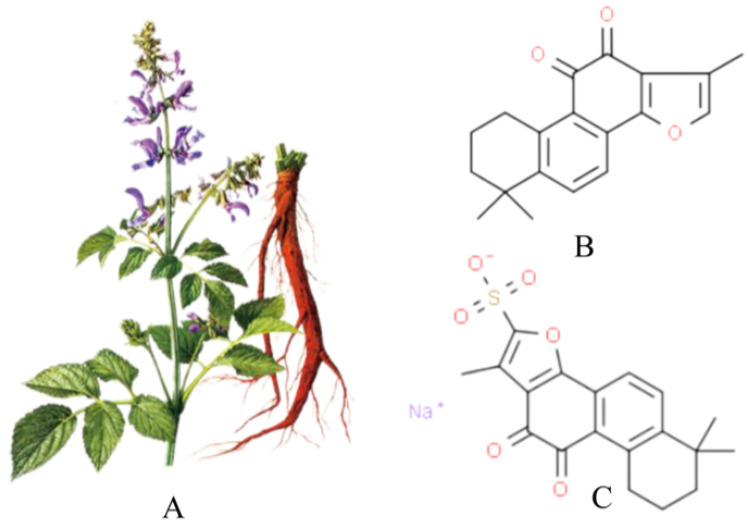
*Salvia miltiorrhiza*, Tan IIA and STS. (**A**) is the picture of *salvia miltiorrhiza*; (**B**) is tanshinone IIA; (**C**) is STS.

**Figure 2 molecules-28-01990-f002:**
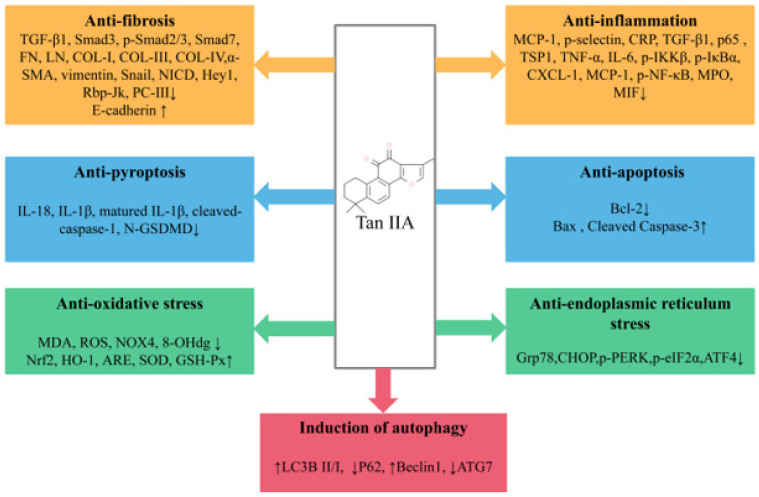
Pharmacological role of Tan IIA in the treatment of kidney disease.

## Data Availability

The original data used in this paper are all from published papers and can be obtained from references in the text.

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
