# Peer review of "Renoprotective Effects of Tanshinone IIA: A Literature Review"

_molecules, 2023, doi:10.3390/molecules28041990_

Round 1

Reviewer 1 Report

The paper needs minor revision.

Corrections are highlighted.

Scientific names should be in italics

Author Response

Dear Reviewer, Thank you very much for your detailed review comments. I have revised my manuscript according to your comments, including fixing the case of the initial letter of the English language and changing the font of the academic name to italic, etc.

Reviewer 2 Report

This is an uncomplicated review focused on the renoprotective effects of tanshinone IIA. The overall level of the manuscript is average. I would like to advise the authors to look through the manuscript and double-check the English language, grammar, and format, because now there are many mistakes and unclear sentences. There are few literatures directly related to the renal protective effect of tanshinone IIA. Relevant documents are not reasonably quoted in some parts. In addition, the format of references is also very confusing. The structure of Fig. 1 and Fig. 2 is not clear and the resolution is low, and the layout of Fig. 2 is not reasonable and beautiful. Therefore, I recommend rejecting it.

Author Response

Point 1:This is an uncomplicated review focused on the renoprotective effects of tanshinone IIA. The overall level of the manuscript is average.

Response 1:Dear reviewers, this article is indeed an uncomplicated review, but it is also the first review article that describes tanshinone IIA in the treatment of kidney diseases, and summarizes the current research status and future development prospects of tanshinone IIA in kidney diseases. In addition, this article has been revised to a large extent according to the comments of you and other reviewers. I believe it will be helpful to improve the level of the manuscript.

Point 2:I would like to advise the authors to look through the manuscript and double-check the English language, grammar, and format, because now there are many mistakes and unclear sentences

Response 2:According to your opinions, I have adopted the language polishing service of MDPI , and I have repeatedly proofread the language, grammar and format of the paper to ensure that the language expression is smooth and accurate.(Please check the attachment for the language polishing certificate)

Point 3:There are few literatures directly related to the renal protective effect of tanshinone IIA

Response 3:Previously, we mainly searched the literature between January 2012 and October 2022, which may lead to the incompleteness of our included literature. Based on your valuable comments and to ensure the completeness of the review, we re-searched the relevant literature, this time setting the search time to all articles published before February 10, 2023. Our search strategy was designed according to the characteristics of different databases. We searched PubMed, EMBASE and CNKI with "tanshinone IIA" and "kidney disease" as keywords up to February 10, 2023.We use pubmed as an example to show the relevant search formula: (((((kidney[Title/Abstract]) OR (renal[Title/Abstract])) OR (nephro[Title/Abstract])) OR (nephritis[Title/Abstract]) ) OR (nephropathy[Title/Abstract])) AND ((((Tanshinone IIA[Title/Abstract]) OR (Sodium tanshinone IIA sulfonate[Title/Abstract])) OR (STS[ Title/Abstract])) OR (Tan IIA [Title/Abstract])). Among them: 673 papers were retrieved from pubmed, 63 papers were retrieved from CNKI (we selected core journals to ensure the quality of the articles) 1520 papers were retrieved from embase, then we removed duplicates, then we browsed by title, abstract and even full text to remove irrelevant papers and papers of poor quality. A total of 32 papers were included in the review, an increase of 7 papers (mainly new publications) compared to the original.

Point 4: Relevant documents are not reasonably quoted in some parts. In addition, the format of references is also very confusing

Response 4:I have checked each documents to ensure that the corresponding citations are reasonably accurate, and some unreasonable applications have been deleted in the article. Regarding the reference format, I have carefully revised it according to the requirements of the journal

Point 4:The structure of Fig. 1 and Fig. 2 is not clear and the resolution is low, and the layout of Fig. 2 is not reasonable and beautiful. 

Response 5:I have redrawn them by referring to the recently published literature on molecules, trying to make them clearer in structure and more logical in layout, as well as prettier. Also, I have uploaded high resolution images to the journal.

Finally, thank you very much for your comments, your valuable advice also makes this article more perfect, wish you a happy work and life.

Reviewer 3 Report

The manuscript written by Chen et al on the “Renoprotective effects of tanshinone IIA: a literature review” can be a promising contribution to the literature. The manuscript can be accepted after suggested major corrections.

1.    There must be a space between word and abbreviation like nephropathy (DN), renal fibrosis (RF) and keep the same pattern throughout the manuscript.

Replace natural products with natural product as only one natural product (tanshinone IIA) is mention in the title.

2.    The authors mentioned the name of the compounds tanshinone IIA thorugh out the manuscript and repeatedly used. Better to use the name once with abbreviation (Tan IIA) and replace the name with abbreviation.

3.    The authors did not take care of the space between comma and words i.e. in the author’s list, there must be space between comma and names.

4.    The authors mention headings 2.2. first and 2.1. and then 2.2. and 2.3 which is confusing for the readers.

5.    I suggest to add these references introduction part related to the tanshinone IIA. 

Andersen S, Tarnow L, Rossing P, Hansen BV, Parving HH. Renoprotective effects of angiotensin II receptor blockade in type 1 diabetic patients with diabetic nephropathy. Kidney international. 2000 Oct 1;57(2):601-6.

Fang ZY, Zhang M, Liu JN, Zhao X, Zhang YQ, Fang L. Tanshinone IIA: a review of its anticancer effects. Frontiers in Pharmacology. 2021 Jan 14;11:611087.

Shang Q, Xu H, Huang L. Tanshinone IIA: a promising natural cardioprotective agent. Evidence-Based Complementary and Alternative Medicine. 2012 Oct;2012.

6.    There are some spaces in the table 1 which is better to remove

7.    The image in Figure 1 and 2 is not clear. The size of the text in both the figures is very small and unclear to the readers. 

8.    The heading and sub headings in the text are not according to the style and format of the journal.

9.    Some of the references in the reference section are not according to the journal style which need to carefully check before the re submission.

Author Response

Point 1: There must be a space between word and abbreviation like nephropathy (DN), renal fibrosis (RF) and keep the same pattern throughout the manuscript.Replace natural products with natural product as only one natural product (tanshinone IIA) is mention in the title.

Response 1: I have revised the article according to your valuable comments.

Point 2: The authors mentioned the name of the compounds tanshinone IIA thorugh out the manuscript and repeatedly used. Better to use the name once with abbreviation (Tan IIA) and replace the name with abbreviation.

Response 2:It has been revised according to your comments

Point 3: The authors did not take care of the space between comma and words i.e. in the author’s list, there must be space between comma and names.

Response 3:It has been revised according to your comments

Point 4: The authors mention headings 2.2. first and 2.1. and then 2.2. and 2.3 which is confusing for the readers.

Response 4:Dear Reviewer, Due to my careless writing, the title of "Renal protective effects of Tan IIA" should be "2.", which has been carefully revised in the article.

Point 5: I suggest to add these references in introduction part related to the tanshinone IIA. 

Andersen S, Tarnow L, Rossing P, Hansen BV, Parving HH. Renoprotective effects of angiotensin II receptor blockade in type 1 diabetic patients with diabetic nephropathy. Kidney international. 2000 Oct 1;57(2):601-6.

Fang ZY, Zhang M, Liu JN, Zhao X, Zhang YQ, Fang L. Tanshinone IIA: a review of its anticancer effects. Frontiers in Pharmacology. 2021 Jan 14;11:611087.

Shang Q, Xu H, Huang L. Tanshinone IIA: a promising natural cardioprotective agent. Evidence-Based Complementary and Alternative Medicine. 2012 Oct;2012.

Response 5:Dear Reviewer, Thank you very much for your opinion, and I have cited the publications you listed in the introduction of this paper.

Point 6:There are some spaces in the table 1 which is better to remove

Response 6:It has been revised according to your comments

Point 7:The image in Figure 1 and 2 is not clear. The size of the text in both the figures is very small and unclear to the readers

Response 7: Dear reviewer, I have redrawn the image, enlarged the related font, and uploaded the related original image to the journal office.

Point 8:The heading and sub headings in the text are not according to the style and format of the journal.

Response 8:Dear reviewer, I have revised the title and subtitle of the main text in accordance with the journal style.

Point 9:Some of the references in the reference section are not according to the journal style which need to carefully check before the re submission.

Response 9:Dear reviewer, I have carefully revised the reference format in accordance with the journal style

Thank you very much for your comments, your valuable comments make this article more perfect, I wish you a happy life.

Round 2

Reviewer 2 Report

The article can be accepted for publication in Molecules.

Reviewer 3 Report

The authors addressed all the comments satisfactory. The manuscript can be accepted in the current format.